# A Portable Biosensor Based on Au Nanoflower Interface Combined with Electrochemical Immunochromatography for POC Detection of Prostate-Specific Antigen

**DOI:** 10.3390/bios12050259

**Published:** 2022-04-19

**Authors:** Yanzhi Dou, Zhenhua Li, Jing Su, Shiping Song

**Affiliations:** 1Division of Physical Biology, CAS Key Laboratory of Interfacial Physics and Technology, Shanghai Institute of Applied Physics, Chinese Academy of Sciences, Shanghai 201800, China; douyanzhi@sinap.ac.cn (Y.D.); lzh@sinap.ac.cn (Z.L.); 2University of Chinese Academy of Sciences, Beijing 100049, China; 3State Key Laboratory of Oncogenes and Related Genes, Institute for Personalized Medicine, School of Biomedical Engineering, Shanghai Jiao Tong University, Shanghai 200030, China; sujing@sjtu.edu.cn; 4The Interdisciplinary Research Center, Shanghai Synchrotron Radiation Facility, Zhangjiang Laboratory, Shanghai Advanced Research Institute, Chinese Academy of Sciences, Shanghai 201210, China

**Keywords:** portable biosensor, Au NFs interface, EIC, POC detection, PSA

## Abstract

Serum prostate-specific antigen (PSA) is a widely used for the detection of prostate cancer and is considered the most reliable biomarker. However, the currently reported detection methods cannot achieve rapid monitoring. Here, we report a novel electrochemical immunochromatography (EIC) system for clinically accurate PSA detection. First, we constructed a carbon interface modified with gold nanoflowers (Au NFs) based on screen-printed carbon electrodes (SPCE), which acted as nanostructures with larger specific surface area that increased the number of PSA capture antibodies and can further improve detection signal-to-noise (S/N) ratio. Then, we fabricated detection chips by combining the SPCE/Au NFs with EIC. Under optimized conditions, the proposed biosensor exhibits high accuracy, taking only 15 minutes to complete detection. By measuring the levels of PSA in clinical blood samples, the biosensor can successfully discriminate clinically diagnosed prostate cancer patients from healthy controls.

## 1. Introduction

Prostate cancer is one of the most common male malignancies all over the world; prostate cancer mortality has been rising year after year, and is now the fifth most common cause of death in men [1]. Prostate-specific antigen (PSA) is an approximately 30 kD single-chain serine protease expressed by the human prostate epithelium; it is a widely utilized and considered the most reliable biomarker in human serum for prostate cancer early diagnosis and prognosis treatment [2]. Currently, the methods used for PSA detection include enzyme-linked immunosorbent assay (ELISA) [3,4], surface-enhanced raman scattering (SERS) [5,6,7], chemiluminescence immunoassay (CLIA) [8,9,10,11,12,13], and fluorescent immunoassay (FIA) [14,15,16]. However, these methods have certain limitations regarding early diagnosis and prognostic monitoring, such as large equipment, complex operation, time requirement, and a lack of amenability to automatic detection. Thus, a simple, accurate, and rapid automated inspection system is urgent and necessary.

Electrochemical biosensors based on screen-printed carbon electrodes (SPCEs) have been successfully used for portable and rapid detection due to their cheap and easy-to-integrate characteristics [17,18,19]. However, the SPCE has a rough surface and poor conductivity, which does not permit excellent detection performance for biomarkers. Recently, nanotechnology-based functional DNA and inorganic nanomaterials have been widely used for the construction of electrochemical biosensor interfaces [20,21,22,23,24,25,26,27]. Some nanomaterials have special properties with good electrical conductivity and high specific surface area, such as gold nanoparticles [28,29], carbon nanotubes [22,30], graphene [31,32], and so on. It is precisely because of these characteristics that nanomaterials exhibit special optical, electrical, and biological properties. Therefore, modification of carbon-based interfaces with these nanomaterials can effectively compensate for the above defects. Among them, Au NFs, as nanoscale materials with advantageous electrical and biological properties, are widely used in biosensor research. Li et al. constructed a nano-integrated microfluidic biochip modified with Au NFs for the enzyme-based point-of-care detection of creatinine [18]. Su et al. developed on-electrode synthesis of shape-controlled hierarchical flower-like gold nanostructures for efficient interfacial DNA assembly and sensitive electrochemical sensing of microRNA [29]. The good electrical conductivity, biocompatibility, and high specific surface area of Au NFs can improve the biosensing performance on the SPCE and amplify the signal.

Good integration with detection chips plays a vital role in achieving rapid and automatic detection. Feng et al. reported a DNA tetrahedron-mediated immune-sandwich assay for the rapid and sensitive detection of PSA through a microfluidic electrochemical detection system, with the characteristics of short reaction time and portability [17]. Compared with a microfluidic electrochemical detection chip, not only the immunochromatography (IC) can integrate an entire laboratory (including reaction zones and power drive modules for liquid flow) into a single chip, but for commercial large-scale manufacturing, the fabrication cost of immunochromatographic structures is much lower than the microfluidic structures. [33,34,35]. In this work, we developed a novel PSA electrochemical immunochromatography detection system. This system consists of two parts: (1) the biosensor chip based on an SPCE modified with Au NFs; (2) the IC structure on the upper layer of the biosensor chip. The Au NFs with a high specific surface area was electrodeposited on the SPCE interface, which can provide more adsorption sites for the PSA capture antibody and is beneficial to the effective recognition of PSA. The IC structure integrated onto an electrochemical chip can greatly shorten the reaction time. Our fabricated EIC detection system exhibited good applicability to drive industrial applications toward multi–scenario POCT.

## 2. Materials and Methods

### 2.1. Materials

Bovine serum albumin (BSA), casein, Tween 20, and other chemicals were purchased from Sinopharm Chemical Reagent Co., Ltd., and TMB (3, 3′, 5, 5′ tetramethylbenzidine) substrate (containing H_2_O_2_) was purchased from Neogen (Lexington, KY, USA). PSA, anti-PSA antibody (PSA mAb), and anti–PSA antibody peroxidase conjugate (PSA mAb-HRP) were purchased from Fitzgerald Industries International (Acton, MA, USA). Hydrogen tetrachloroaurate (III) hydrate (HAuCl_4_) was purchased from J&K Scientific Ltd. Normal human sera were obtained from Renji Hospital, School of Medicine, and Shanghai JiaoTong University. The 16–channel screen–printed carbon electrode (16-SPCE) was purchased from Zhejiang Nanosmart Biotechnical Co., Ltd. (Ningbo, China).

The buffer solutions involved in this study are as follows: the antigen was dissolved in 0.01 M phosphate-buffered saline solution (PBS). The coating antibody was dissolved in a 0.01 M sodium carbonate (CB) buffer; the SPCE was pretreated in 0.1M phosphate buffer (PB) solution, and the PBST solution was used to clean the modified SPCE. All chemical reagents were prepared with ultrapure water from a Millipore Milli-Q water purification system (18.2 MΩ cm resistivity).

### 2.2. Construction of PSA Detection Capture Interface

#### 2.2.1. Electrodeposition of Au NFs on the SPCE

The SPCE for PSA detection was rinsed with Milli-Q water and pretreated by cyclic voltammetry (CV) scanning in 0.1 M PB buffer solution in order to clean the surface of the working electrode (WE) (Appendix A), which was dried with N_2_ airflow. Then, the Au NFs were electrodeposited on the working electrode surface of the SPCE through potentiostatic deposition in 50 µL 0.1% HAuCl_4_ solution [18]. The parameters were as follows: deposition potential: −0.2 V, deposition time: 300 s, sample interval: 0.1 s. After the deposition of Au NFs, the electrodes were rinsed clean and ready for use (Appendix A).

#### 2.2.2. Au NFs/SPCE Layer Characterization

The SPCE/Au NFs and the bare SPCE were characterized using a scanning electron microscope (SEM, LEO 1530 VP, Zeiss, Oberkochen, Germany).

#### 2.2.3. Optimization the Concentration of PSA mAb Capture Antibody

The PSA mAb, as capture molecules, was diluted to different concentrations in CB solution: 10 µg/mL, 20 µg/mL, 40 µg/mL, 60 µg/mL, 100 µg/mL, and 200 µg/mL. Then, 10 µL of each different concentration of solution was dropped onto the surface of the SPCE/Au NFs chip and incubated for 16 h at 4 °C. After incubation, the electrode chips were blocked with 1% BSA + 1% casein solution for 2 h at 37 °C, and 10 µL of 100 ng/mL PSA was coated on the electrodes for 1 h at 37 °C. The electrode chips were rinsed repeatedly with PBS and PBST solution, then dried with N_2_ airflow. Subsequently, 10 µL of PSA mAb–HRP was dropped onto the SPCE/Au NFs/PSA mAb-PSA surface for 1 h at 37 °C. The immunoelectrode chip was repeatedly cleaned with PBST and PBS solution. Finally, 50 µL of TMB substrate was dropped onto the surface of three–electrode system. The electrochemical tests were performed on the 16–channel electrochemical workstation.

#### 2.2.4. Preparation of the SPCE/Au NFs/PSA mAb Capture Layer and EIC Detection Chip

The optimal coating concentration of PSA mAb was 100 µg/mL in CB solution; 10 µL was coated on the SPCE/Au NFs chip for 16 h at 4 °C and rinsed three times with PBS. Then, the electrode chips were blocked with 1% BSA + 1% casein solution for 2 h at 37 °C, washed three times with PBST and PBS, respectively, and dried with N_2_ airflow. Then, 10 µg/mL of PSA mAb–HRP was immobilized on the front of the sample pad and allowed to dry. We pasted the sample pad, NC film, and absorbent pad on the SPCE prepared above to prepare an EIC detection chip.

#### 2.2.5. Electrochemical Analysis of the PSA with EIC Detection Chip

All electrochemical tests were performed with a smartphone-based electrochemical device. A high concentration of PSA was diluted to different concentrations, from 0 ng/mL to 300 ng/mL, with 10 mM PBS solution and normal human serum, successively. Then, 100 µL of each solution at each concentration was dropped into the sample well of EIC detection chip. After 15 min, 100 µL of PBST washing solution was dropped into the sample well for 5 min. Finally, 50 µL of TMB substrate was added in the detection well, the current signal for PSA detection was tested by cyclic voltammetry (CV) for 2 segments in the potential window of −0.3 V to 0.6 V and for amperometric (i–t) measurement at the potential of −0.1 V for 30 s. We used CV and i–t measurements to test the current response in subsequent experiments.

#### 2.2.6. Clinical Sample Test with EIC Biosensor Chip

The serum samples from 19 healthy individuals were selected as the control, and samples from 27 prostate cancer patients were tested. The clinical study was approved by the Ethics Committee. For analysis, 100 μL of each patient sample was used.

## 3. Results and Discussion

### 3.1. Design of the Smartphone-Based EIC Biosensor for PSA Detection

The principle of the smartphone-based EIC biosensor chip for PSA detection was illustrated (Figure 1). The EIC biosensor system is composed of two parts: (1) the smartphone-based electrochemical detector; (2) the EIC biosensor chip. The SPCE for constructing the biosensor is composed of a carbon working electrode (WE), an Ag/AgCl reference electrode (RE), and a carbon counter electrode (CE). We electrodeposited Au NFs on the WE interface to improve the conductivity of electrodes, and the high specific surface area of Au NFs can provide more adsorption sites for PSA capture antibodies. Then, the PSA mAb as capture molecules was modified on the SPCE/Au NFs by physical adsorption for specific recognition of PSA molecules. The SPCE/Au NFs/PSA mAb chip was integrated with the upper sample pad, NC film, and absorbent pad to form the EIC structure. The HRP–labeled PSA mAb were immobilized on the front of the sample pad. A card case with a sample hole and a detection hole was installed outside the EIC chip. The PSA biomarker was added to the sample hole and combined with PSA mAb-HRP to form the PSA mAb–HRP/PSA complex, which was followed by the chromatographic direction to the SPCE/Au NFs/PSA mAb capture layer and binding to the PSA capture antibody, finally forming PSA mAb/PSA/PSA mAb–HRP double-body sandwich complexes on the surface of SPCE/Au NFs. The i–t measurement was employed for the determination of PSA. The reduction current signal was generated by HRP catalyzed TMB substrate added in the detection well, which was positively correlated with the concentration of PSA.

### 3.2. Structural Characterization of the SPCE/Au NFs and Its Performance

To confirm the formation of 3D Au NFs structure on the SPCE interface, SEM was used to further investigate the surface topography of the SPCE and SPCE/Au NFs at different magnifications. As shown in Figure 1a,c, the bare SPCE exhibited a very rough surface area. Figure 1b,d showed the nanostructure of electrodeposited Au on the surface like a flower and the size of the Au NFs was about 200 nm–250 nm. The electrochemical properties of the bare SPCE and SPCE/Au NFs layer were demonstrated by CV and EIS. As shown in Figure 1e, the AuNFs deposited on the SPCE significantly improved the response current of the Fe (CN)_6_^3−/4−^ redox probe. Compared with the bare SPCE, the difference in the redox current peak potential of SPCE/AuNFs became smaller, indicating that the AuNFs could accelerate the electron transfer rate of redox probes and improve the electrochemical responses. Figure 1f shows the SPCE/Au NFs layer possessed a smaller charge transfer resistance (Rct) than bare SPCE, indicating that Au NFs could obviously improve the conductivity of the SPCE surface. The PSA mAb was modified on the SPCE/Au NFs by a physical adsorption method, Rct increased, which indicated that the PSA mAb layer on the SPCE surface inhibited the charge transfer rate of the redox probe.

### 3.3. Optimization of Experimental Conditions

To improve the detection performance of the PSA EIC biosensor chip, some key parameters were optimized, including the PSA capture antibody concentration and chromatographic reaction time. First, we optimized the capture antibody coating concentration in the detection system. As shown in Figure 2a, with the concentrations of PSA capture antibody increased from 10 µg/mL to 200 µg/mL on the surface of electrode, the current signal by i–t measurement and the S/N ratio increased. The highest signal was observed at the coating antibody concentration of 100 µg/mL, which showed that sufficient antibodies were immobilized on the electrode surface. The chromatographic reaction time was an important parameter in the EIC detection system. The shorter the reaction time, the better it can meet the needs of early diagnosis and prognostic detection of prostate cancer patients. We chose chromatographic reaction times from 6 min to 20 min. Figure 2b shows that the current values and S/N were higher when the time was 15 min. When the chromatographic reaction time increased to 20 min, the current signal and S/N ratio decreased, which was due to the decreased concentration of PSA/PSA mAb–HRP complex above the electrode, and decreased the binding efficiency with the capture antibody on the surface of the SPCE/Au NFS. At the same time, the prolongation of the chromatographic reaction time led to increased amounts of PSA mAb–HRP adsorption on the WE surface, which generated a high background current signal, leading to a reduced S/N ratio. The best chromatographic reaction time was 15 minutes, which was selected for subsequent studies.

### 3.4. Evaluation Performance of EIC Biosensing Chip of PSA Based on the Au NFs Interface

We employed the EIC biosensing chip to detect the prostate cancer biomarker (PSA) by modifying the SPCE/Au NFs with the recognition element that specifically captured each biomarker. Under the optimized conditions, we evaluated the analytical performance of PSA. As shown in Figure 3a, two pairs of distinct redox peaks were observed, indicating that potentials lower than 0.05 V should be suitable for measurements. We chose −0.1 V for the following i–t measurements (Figure 3b). The standard PSA solution was diluted to different concentrations from 0 to 300 ng/mL. The reduction current generated by the HRP catalyzed substrate TMB increased with the increased concentration of PSA. Figure 3c showed the linear relationship between the reduction current values and the PSA concentration from 0 to 300 ng/mL. The linear regression equation was y = 15.18x + 75.96; the coefficient of determination (R^2^) of the regression line was over 0.99. Clinically, the PSA value of prostate cancer patients is higher than 10 ng/mL. Based on the results of linear regression analysis, the LOD of PSA was 0.28 ng/mL (8.75fM) (y = Blank + 3SD), which could meet the needs of clinical detection.

We further performed detection of prostate cancer biomarker that was spiked in normal human mixed serum. As shown in Figure 3d, the linear regression equation was y = 9.6x + 85, and the coefficient of determination (R^2^) was 0.99. The value of the LOD was 1.84 ng/mL (57.5 fM). The biosensor chips accurately discriminated the single biomarker down to femtomolar concentrations. Compared with the PBS buffer solution, the LOD is reduced in the pooled blood serum, due to the steric hindrance of the existence of hundreds of non–target proteins in the serum causing a decrease in the specific recognition efficiency of PSA and PSA capture antibody. The detection levels of the PSA in the normal human mixed serum could meet the needs of prostate cancer in clinical diagnosis.

To verify the specificity of the EIC biosensor chip, we examined its cross–reaction with other protein molecules, including AFP and CEA present in real samples. The current values were measured at the same concentration levels of the interferents. As shown in Figure 3e, the current response signal to PSA was much higher than those of AFP and CEA, and the signals of AFP and CEA were near to the signal of the blank. No cross-reactivity was produced when testing against the two protein molecules above.

### 3.5. The Performance of EIC Biosensing Chip of Clinical Relevance

To evaluate the biosensor chip’s capability to discriminate prostate cancer patients from healthy controls, we selected 19 healthy persons and 27 prostate cancer patients for testing. As shown in Figure 4a, the current values of the healthy control were lower than prostate cancer patients. In Figure 4b, the box plots show the estimated levels of PSA in the serum. Statistical analysis was performed by means of one–way analysis of variance (ANOVA). *** *p* < 0.001. The results showed that there were significant differences between healthy controls and prostate cancer patients.

## 4. Conclusions

In summary, we developed a novel EIC biosensing system based on Au NFs modifying the SPCE interface for PSA detection. The Au NFs were electrodeposited on the SPCE surface to improve the efficiency of electron transfer and provide more capture antibody adsorption sites of surfaces. We obtained a good linear range from 0 to 100 ng/mL, and the LOD was 0.28 ng/mL (8.78 fM) with a total reaction time of less than 20 min. In addition to its femtomolar sensitivity, the biosensor chips selectively recognized prostate cancer biomarkers in the serum. By detecting PSA biomarker in clinical serum samples, the EIC biosensor chip successfully distinguished prostate cancer patients from healthy controls (*p* < 0.001). Moreover, this EIC system was easy to use without professional skills and complicated instruments. On the basis of this work, we believe that this EIC detection system has huge commercialization potential for the development of POCT devices because of the characteristics of high integration and low manufacturing cost, and has broad application prospects in clinical diagnosis, and therefore holds the potential for greater economic and social value in practical applications.

## Data Availability

Not applicable.

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
