# Peer review of "A Portable Biosensor Based on Au Nanoflower Interface Combined with Electrochemical Immunochromatography for POC Detection of Prostate-Specific Antigen"

_biosensors, 2022, doi:10.3390/bios12050259_

Round 1

Reviewer 1 Report

The paper presents a study on the fabrication and testing of an electrochemical immunochromatography system for PSA detection. The paper can’t be published in the present form. The manuscript contains unproperly used scientific terms. Therefore, I suggest the authors a thoroughly revision of the text in terms of scientific meaning. Some of the changes that need to be made are presented below.

Line 46 – The expression “DNA nanomaterials“ is not clear

Line 53 –„Au NFs are nanoscale with advantages of electrical and biological properties for biosensing researches.“ - The phrase needs to be reformulated

Line 91 – „The SPCE for PSA detection were rinsed with Milli-Q water and pretreated to clean the surface of the three-electrode system by cyclic voltammetry (CV) with the 16-channel electrochemical Work station“ – reformulation; the pre-treatment is not described

Line 96 - The preparation of the AuNFs/SPCE modified electrodes is not clear: „potentiostatic deposition. The parameters were as follows: deposition potential: -0.2 V, deposition time: 300 s, scan rate: 0.1 V/s“

What is the relevance of scan rate when using potentiostatic deposition?

Line 126: „50 µL of TMB substrate were added in detection well by cyclic voltammetry (CV) and amperometric (i-t) measurement.“ The phrase is not clear

Line 155: „The reduction current generated by HRP catalysed the TMB substrate added in the detection hole has a certain positive correlation with the concentration of PSA.“ The working principle of the developed sensor is not clearly evidenced. The expression „has a certain positive correlation with the concentration of PSA“ can’t be accepted as argument in a scientific paper

Line 162: „The electrochemical properties of the bare SPCE and SPCE/Au NFs layer were demonstrated by CV and EIS. As shown in Figure 1e, the bare SPCE displayed a pair of small redox peaks, the SPCE/Au NFs showed a pair of quasi-reversible redox peaks. The result should  be attributed to improving the conductivity of the SPCE surface by Au NFs.“ This paragraph denotes a limited acquaintance with the electrochemistry field. The authors should indicate that these electrochemical signals (redox peaks) are attributed for the Fe(CN)6 3-/4- species in the testing solution

Line 178, section 3.3 – this section is not clearly presented. How were the currents presented in Fig 2 obtained? What was the electrochemical technique employed for testing? Again, the working principle of the sensor is not clearly presented.

Line 202, section 3.3 (the numbering is repeating) In this section one can deduce the electrochemical technique employed in testing (chronoamperometry). However, this should be explained when first introduced in text (the previous section)

Some of the statements needs to be properly reformulated, and also certain conclusions must be drawn after inserting these statements.  I suggest a thoroughly check of the entire article.

Reviewer 2 Report

The manuscript describes a portable biosensor for PSA detection in clinical samples. The manuscript was carefully prepared, logically organized and is easy to follow. The topic is interesting and falls in the readership of Biosensors. Some suggestions are detailed below:

  1. Additional justification about the novelty of this study with respect to other PSA sensors is recommended. Perhaps specific differences would help.
  2. Physical adsorption method for Ab immobilization at AuNFs-modified electrode was employed. Is it sufficient enough not to get washed during the immunocromatographic reaction?
  3. There are extensive researches carried out for the development of distinct sensing platforms like electrochemical, microfluidic etc for PSA. These sensors are sensitive, rapid, and low cost. How do you justify your sensor as better than these?
  4. The authors need to discuss the market value of this type of sensor. What is the future of this device in terms of commercialization and economic value?

Minor errors

  • What do the “error bars” stand for in figure caption 1?

Round 2

Reviewer 1 Report

The manuscript has been reviewed and improved. However there are still a few aspects that should be clarified before publication.

  1. I am still recommending the authors to reanalyse the paragraph at Line 96 - The preparation of the AuNFs/SPCE modified electrodes is not clear: “potentiostatic deposition. The parameters were as follows: deposition potential: -0.2 V, deposition time: 300 s, scan rate: 0.1 V/s”. The authors are mentioning the employment of potentiostatic deposition, but in the same time they are providing a scan rate. What is the meaning of this scan rate?
  2. The working principle of the developed sensor is schematically illustrated in Scheme 1. The authors should provide a short description of this working principle in the caption of the figure.
